# Changes in the Mitochondrial Dynamics and Functions Together with the mRNA/miRNA Network in the Heart Tissue Contribute to Hypoxia Adaptation in Tibetan Sheep

**DOI:** 10.3390/ani12050583

**Published:** 2022-02-25

**Authors:** Yuliang Wen, Shaobin Li, Fangfang Zhao, Jiqing Wang, Xiu Liu, Jiang Hu, Gaoliang Bao, Yuzhu Luo

**Affiliations:** Gansu Key Laboratory of Herbivorous Animal Biotechnology, Faculty of Animal Science and Technology, Gansu Agricultural University, Lanzhou 730070, China; yuliang_wen@163.com (Y.W.); zhaofangfang@gsau.edu.cn (F.Z.); wangjq@gsau.edu.cn (J.W.); liuxiu@gsau.edu.cn (X.L.); huj@gsau.edu.cn (J.H.); baogl@st.gsau.edu.cn (G.B.)

**Keywords:** hypoxia, RNA-seq, mitochondria, dynamics, oxidative phosphorylation, Tibetan sheep, Qinghai–Tibet Plateau

## Abstract

**Simple Summary:**

Long-term exposure to hypoxia, a major source of cellular stress, can induce hypoxia-related diseases and even death. Mitochondria play an important role in mediating the energy production response to hypoxia, but little is known about the mechanisms involved. Tibetan sheep are mainly distributed across the Qinghai–Tibet Plateau, where they have adapted well to hypoxia after long-term adaptation. In this work, a systematic analysis of the blood indexes, tissue morphology, mRNA and miRNA expression regulation, and changes in the mitochondrial function of Tibetan sheep at different altitudes was carried out to provide insights into the mechanism of animal adaptation to hypoxia and the progression of hypoxia-related illness.

**Abstract:**

This study aimed to provide insights into molecular regulation and mitochondrial functionality under hypoxia by exploring the mechanism of adaptation to hypoxia, blood indexes, tissue morphology, mRNA/miRNA regulation, mitochondrial dynamics, and functional changes in Tibetan sheep raised at different altitudes. With regard to blood indexes and myocardial morphology, the HGB, HCT, CK, CK-MB, LDH, LDH1, SOD, GPX, LDL level, and myocardial capillary density were significantly increased in the sheep at higher altitudes (*p* < 0.05). The RNA-seq results suggested the DEmRNAs and DEmiRNAs are mainly associated with the PI3K-Akt, Wnt, and PPAR signaling pathways and with an upregulation of oncogenes (*CCKBR*, *GSTT1*, *ARID5B*) and tumor suppressor factors (*TPT1*, *EXTL1*, *ITPRIP*) to enhance the cellular metabolism and increased ATP production. Analyzing mRNA–miRNA coregulation indicated the mitochondrial dynamics and functions to be significantly enriched. By analyzing mitochondrial dynamics, mitochondrial fusion was shown to be significantly increased and fission significantly decreased in the heart with increasing altitude (*p* < 0.05). There was a significant increase in the density of the mitochondria, and a significant decrease in the average area, aspect ratio, number, and width of single mitochondrial cristae with increasing altitudes (*p* < 0.05). There was a significant increase in the NADH, NAD+ and ATP content, NADH/NAD+ ratio, and CO activity, while there was a significant decrease in SDH and CA activity in various tissues with increasing altitudes (*p* < 0.05). Accordingly, changes in the blood indexes and myocardial morphology of the Tibetan sheep were found to improve the efficiency of hemoglobin-carrying oxygen and reduce oxidative stress. The high expression of oncogenes and tumor suppressor factors might facilitate cell division and energy exchange, as was evident from enhanced mitochondrial fission and OXPHOS expression; however, it reduced the fusion and TCA cycle for the further rapid production of ATP in adaptation to hypoxia stress. This systematic study has for the first time delineated the mechanism of hypoxia adaptation in the heart of Tibetan sheep, which is significant for improving the ability of the mammals to adapt to hypoxia and for studying the dynamic regulation of mitochondria during hypoxia conditions.

## 1. Introduction

Oxygen is necessary for normal aerobic metabolism in mammals, mainly during aerobic metabolism in mitochondria for producing adenosine triphosphate (ATP) in order to maintain the normal physiological functions of the cells. Hypoxia is a common natural phenomenon in nature. The oxygen concentration gradually decreases with an increase in altitude, and the oxygen content at an altitude of 4000 m is only 60% of that at sea level [1]. A long-term exposure to plateau hypoxia causes an insufficient supply of oxygen to cells, resulting in tissue injury and even death of the animals. Meanwhile, hypoxia is more common in the development of cancer cells. Due to malignant proliferation, the cancer cells compete with each other for oxygen, leading to a hypoxic microenvironment. Nowadays, many studies of anticancer drugs have targeted the energy metabolism of cancer cells [2]. Therefore, studying the mechanism of hypoxic adaptation is highly significant for improving both the ability of animals to adapt to hypoxia and cancer-related medical research.

Long-term exposure to hypoxia induces a variety of acute mountain sicknesses (AMSs) characterized by erythrocytosis and pulmonary edema. However, long-term adaptive evolution in plateau-indigenous mammals such as Tibetan sheep [3] and Tibetan pigs [4] has ensured their systematic adaptation to plateau hypoxia in terms of their morphological structure, physiology, biochemistry, and genetics. Tibetan sheep are mainly distributed across the Qinghai–Tibet Plateau and its adjacent areas at an altitude of 2500–4500 m. They are reportedly derived from wild sheep after long-term adaptation, with their population estimated to exceed 50 million. Moreover, these sheep are not only well-adapted to the plateau hypoxia, but also provide meat, wool, and milk for local herdsmen, serving as one of their main local economic resources [3,5]. These traits make Tibetan sheep an ideal model organism for studying hypoxic adaptation.

In recent years, transcriptomics has been reported to have an important role in revealing many cellular events. Transcriptional analysis reveals the differential expression of hypoxia regulatory factors that are required to adapt to a hypoxia environment [6]. Hypoxia-inducible factor 1 (HIF-1), phosphoinositide 3-kinase-Akt (PI3K-Akt), and peroxisome proliferator-activated receptor (PPAR) signaling pathways [7,8] have been reported to be closely related to hypoxia, and the miRNAs such as miR-21-5p [9], miR-200b/C [10], and miR-21 [11] have been shown to play an important role in regulating hypoxia.

Changes in glucose metabolism and glycolysis are up-regulated for producing ATP to resist hypoxia stress [12]. A mitochondrion is a double-layered membrane organelle possessing its genome and transcription system. It can produce most of the ATP (95%) required by cells using the tricarboxylic acid (TCA) cycle and oxidative phosphorylation (OXPHOS) to meet the energy requirement of the cells [13]. Meanwhile, mitochondria also regulate the production and storage of reactive oxygen species (ROS), calcium homeostasis, and apoptosis [14,15]. Mitochondrial dynamics is an important aspect of the adaptation of the cells to hypoxia. The rate of fusion and fission is regulated by key proteins such as optic atrophy 1 (OPA1), dynamin-related protein 1 (Drp1), and fission 1 (Fis1) [16,17]. Mitochondrial fusion and fission restrain each other under physiological conditions, such that mitochondria reach a certain dynamic balance for supplementing and repairing mitochondrial DNA (mtDNA), regulating the TCA cycle, and OXPHOS, providing energy to cells and meeting the energy needs of the cells [18]. Disrupting this balance, i.e., when mitochondrial fusion and fission are blocked, results in a variety of diseases [19].

The present research on hypoxia adaptation mainly focuses on blood indexes, histomorphology, and gene expression, and only a few studies exist on hypoxia adaptation through energy transport with mitochondria as the starting point. Therefore, in this study, the Tibetan sheep used as a model were taken to different altitudes (different oxygen concentrations), and the expression profile of the heart mRNA and miRNA were analyzed, combining mitochondrial dynamics and function, to explore the mechanism of hypoxia adaptation, thus providing a reference for developing animal husbandry and preventing and treating illnesses related to hypoxia.

## 2. Materials and Methods

### 2.1. Ethics Statement

All the experiments on the sheep were conducted according to the guidelines for the care and use of experimental animals, as established by the Ministry of Science and Technology of the People’s Republic of China (Approval number 2006-398). The study was reviewed and approved by the Faculty Animal Policy and Welfare Committee of Gansu Agricultural University (Ethic approval file No. GSAU-Eth-AST-2021-001).

### 2.2. Sample Collection

In this study, 12 Tibetan sheep ewes were randomly selected according to their distributions, including four ewes at an altitude of approximately 2500 m from Zhuoni county (Gannan, Gansu, China) (TS25), four ewes at an altitude of approximately 3500 m from Haiyan county (Haibei, Qinghai, China) (TS35), and four ewes at an altitude of approximately 4500 m from Zhiduo county (Yushu, Qinghai, China) (TS45). All the ewes were approximately 3.5 years old, healthy, and sheltered in an outdoor paddock with clean water available ad libitum. As the rates of AMS in people increase rapidly from 30 years of age, the age of the sheep investigated was chosen to represent this age group in people. Blood from the jugular vein was collected from 5 mL ordinary and EDTA anticoagulant tubes for determining the blood indices. After sacrificing the ewes, the heart tissue (left ventricle) was collected and snap-frozen in liquid nitrogen for mRNA and miRNA sequencing.

### 2.3. Measurement of Blood Physiological and Biochemical Indexes

Blood was drawn and collected from the jugular vein from each sheep using 5 mL sodium heparin and common tubes, and the blood gas indices, including partial pressure of oxygen (PO_2_), oxygen saturation (SO_2_), hemoglobin (HGB), hematocrit (HCT), hydrogen ion concentration (PH), carbon dioxide pressure (PCO_2_), bicarbonate concentration of bicarbonate (HCO^3−^), and base excess (BE), were measured by an i-STAT blood gas analyzer (Abbott, Chicago, IL, USA). Biochemical indices, including creatine kinase (CK), creatine kinase isoenzymes (CK-MB), lactate dehydrogenase (LDH), lactate dehydrogenase isoenzymes (LDH1), superoxide dismutase (SOD), glutathione peroxidase (GPX), and low-density lipoprotein (LDL), were measured using an automatic biochemical analyzer (Chemray 800, Shenzhen Lei Du Life Technology Co., Ltd., Shenzhen, China). There were three biological repeats and two technical replicates in each index, taking the TS25 Tibetan sheep as the control, and the results were analyzed using Excel 2016 (Office, Microsoft Corporation, Washington DC, USA).

### 2.4. Hematoxylin and Eosin Staining

The left ventricle of the heart was stained with H&E and observed under a microscope (Nikon Eclipse E100, Tokyo, Japan). There were three biological replicates in each group and fifteen microscopic fields in each replicate, while the hearts of the TS25 Tibetan sheep were considered as the control, and the results were analyzed using the Image-Pro Plus 6.0 software (Media Cybernetics Inc., Rockville, MD, USA).

### 2.5. RNA Extraction

The total RNA was extracted from the heart, liver, lung, brain, and quadriceps femoris using a TRIZOL reagent kit (Invitrogen, Carlsbad, CA, USA). The total RNA isolated from the heart was enriched by Oligo dT to remove the rRNA. The quality of the RNA and the samples (RNA integrity number (RIN) > 7) were evaluated using the 1% agarose gel electrophoresis and Agilent 2100 (Agilent Technologies, Palo Alto, CA, USA) for the subsequent tests.

### 2.6. Library Construction and Sequencing for mRNA and miRNA

The library construction was conducted according to the methods as described by Yang et al. (2021) [20], and then the Illumina hiseqtm 2500 (gene denovo Biotechnology Co., Ltd., Guangzhou, China) was used for sequencing. The original mRNA sequencing data (PRJNA780601) and miRNA sequencing data (PRJNA781660) were submitted to the SRA (sequence read archive) database of the NCBI.

### 2.7. mRNA Identification

Principal component analysis (PCA) was performed to confirm the reproducibility of each sample based on the mRNA expression in each sample (Appendix A). Fastp software [21] was used for to obtain the high-quality clean reads necessary for subsequent bioinformatics analysis. Bowtie2 (v2.2.8) [22] was used for comparing the clean reads to the ribosomal RNA (rRNA) database. An index of the reference genome was built, and paired-end clean reads were mapped to the Ovis aries RefSeq (Ovis aries_v1.0) using HISAT 2 (v2.1.0) [23]. Stringtie [24] and RSEM software [25] was used to reconstruct the transcripts and calculate the fragment per kilobase of transcript per million mapped reads (FPKM) values of all the genes in each sample for quantifying the gene expression abundance [26]. Finally, the differentially expressed mRNAs (DEmRNAs) between the groups were analyzed using DESeq 2 software [27].

### 2.8. miRNA Identification

The clean reads were obtained after filtering the raw reads and were aligned with the small RNAs in the GenBank and Rfam database [28,29] to remove rRNA, scRNA, snoRNA, snRNA, and tRNA. All the clean reads were also aligned with the reference genome and searched against the miRbase database [30] to identify the known (*Ovis aries*) miRNAs. The uncommented reads were aligned with the reference genome (Ovis aries_v1.0) by HISAT2 (v.2.4.). The novel miRNAs were identified according to the position of the miRNA genome, the hairpin structures were predicted by mirdeep2, and the transcripts per million (TPM) of each miRNA were calculated for standardizing the expression.

### 2.9. Functional Annotation of DEmRNAs

The gene ontology (GO) and Kyoto encyclopedia of genes and genes (KEGG) pathways of the DEmRNAs between the different groups were annotated using the database for annotation and visualization, and the integrated discovery (DAVID) online analysis tool was used for exploring their location, function, and biological signal pathways [31]. The GO terms and pathways with *q* < 0.05 were considered significantly enriched by the DEmRNAs.

### 2.10. Association Analysis of DEmRNAs-DEmiRNAs

The mRNAs with a fold change ≥1.5 and a false discovery rate (FDR) < 0.05 were identified as DEmRNAs, and the miRNAs with a fold change ≥1.5 and *p* < 0.05 were identified as DEmiRNAs using edgeR (http://www.bioconductor.org/packages/release/bioc/html/edgeR.html, accessed on 22 December 2021). The RNA hybrid (v2.1.2) + svm_light (v6.01), Miranda (v3.3a), and Target Scan (v7.0) were used for predicting the potential target genes of DEmiRNAs, and the genes at the intersection of the results from the three software packages were selected as the predicted miRNA target genes. Due to the potential negative regulatory relationship between mRNAs and miRNAs, the correlation between miRNAs and their predicted target gene expression was evaluated by the Pearson correlation coefficient (PCC). Then, the mRNA and miRNA relationship pairs were screened by PCC < −0.7 and *p* < 0.05, before the mRNA–miRNA network was constructed using Cytoscape (v3.6.0). The online analysis tool Davis was used to annotate the GO and KEGG pathways of the DEmRNAs in the network. Through analysis, mRNAs in the network have been found to possess a biological role mainly by regulating mitochondrial dynamics and function in hypoxia.

### 2.11. qPCR Verification

The total RNA was isolated from the heart tissue using the Trizol reagent and reverse-transcribed into cDNA using Evo M-MLV RT Kit and miRNA 1st strand cDNA (AG, Changsha, China). The SYBR^®^ Green Premix Taq (AG, Changsha, China) was used for qPCR analysis. In total, 20 DEmRNAs and 10 DEmiRNAs were randomly selected for determining the sequencing accuracy. The primers used here were designed using the Primer 5.0 software, synthesized by Beijing AuGCT DNA-SYN Biotechnology Co., Ltd. (Beijing, China), and the mRNA primers and miRNA primers are listed in Appendix A, respectively. The ribosomal protein L19 (*RPL19*, accession number: XM_012186026.3) and *β-actin* (accession number: NM_001009784.3) were used as the internal control of DEmRNAs, and *U6* and *18S* were used as the internal of DEmiRNAs for calculating the relative expression according to the 2^−ΔΔCt^ method [32].

### 2.12. Expression Analysis of Mitochondrial Dynamic-Related Genes

The expression patterns of mitochondrial fusion genes (*OPA1*, *Mic60*) and mitochondrial fission genes (*Drp1*, *MFF*, and *Fis1*) in the heart (left ventricle), liver (left lobe), lung (left lobe), brain (parietal lobe), and quadriceps femoris (left lateral muscle) of Tibetan sheep at different altitudes were studied using qPCR. The total RNA was isolated from five tissues using a Trizol reagent and the primers used were designed using the Primer 5.0 software, synthesized by Beijing AuGCT DNA-SYN Biotechnology Co., Ltd. (Beijing, China) and are listed in Table 1.

### 2.13. Analysis of the Expression of Mitochondrial Dynamic-Related Proteins

Five tissues were collected for immunohistochemical analysis, using the specific methods as described by He et al. [3]. The rabbit polyclonal anti OPA1 (bs-11764R, 1:100; Bioss, Beijing, China), Mic60 (bs-1824R, 1:100; Bioss, Beijing, China), MFF (bs-7628R, 1:100; Bioss, Beijing, China), Drp1 (bs-4100R, 1:100; Bioss, Beijing, China), and Fis1 (bs-7646R, 1:100; Bioss, Beijing, China) were added to the sections and incubated at 4 °C overnight, then incubated with goat anti-rabbit secondary antibody IgG/horseradish peroxide (HRP) (GB23303, 1:200; Servicebio, Wuhan, China) for 50 min at 25 °C. The sections were immunostained and counterstained using DAB and hematoxylin, and the images were captured under the microscope (Nikon DS-U3, Nikon, Tokyo, Japan). Considering the TS25 Tibetan sheep heart integrated optical density (IOD) as the control, each tissue was found to possess three biological repetitions and each biological repetition had six microscopic fields, and the IOD of each tissue was measured by Image-Pro Plus 6.0.

### 2.14. Ultrastructural Observation of the Mitochondria

For TEM (HT7800, HITACHI, Japan) analysis, five tissues were collected in glutaraldehyde (3%) where the samples were treated according to the methods as described by Wang et al. [16]. There were three biological replicates in each group and fifteen microscopic fields in each replicate. Taking the heart of the TS25 Tibetan sheep as the control, the mitochondria dynamic-related indexes, including the density, area, the aspect ratio of mitochondria per unit area, as well as the average number and width of the cristae in a single mitochondrion were measured using the software Image Pro-plus 6.0.

### 2.15. Mitochondrial Functional Analysis

The content of CA, NADH, NAD+, ATP, the activity of SDH, and CO were measured using an assay kit (Nanjing Jiancheng, Nanjing, China) according to the manufacturer’s instructions. The corresponding absorbance was measured using a plate reader (Multiskan FC; Thermo Fisher Scientific, Beijing, China). Three biological repeats and two technical replicates were performed in each index.

### 2.16. Statistical Analysis

All the graphs were generated using GraphPad Prism 8.0 (GraphPad Software Inc, San Diego, CA, USA). Statistical analyses included the analysis of variance (ANOVA), followed by Fisher’s least significant difference test for multiple comparisons in SPSS 20.0 software (IBM, Armonk, NY, USA). The analysis of post-hoc power in Minitab software (v18, Minitab Inc, PA, USA) was performed, and the average power of the test was 0.92. All the experimental data were presented as mean ± standard error of the mean (SEM), where *p* < 0.05 and different lowercase letters indicated that the difference was significant.

## 3. Results

### 3.1. Analyses of Blood Physiological and Biochemical Indices

As shown in Table 2, PO_2_ and SO_2_ were lower in the TS45 Tibetan sheep than in the TS25 sheep (*p* < 0.05). The HGB and HCT contents were higher in the TS45 than in the TS35 and TS25 sheep (*p* < 0.05). The PCO_2_ decreased with an increase in altitude (*p* < 0.05), while BE showed an increase and HCO^3−^ a decrease amongst the biochemical indices. The CK, CK-MB, LDH, LDH1, SOD, GPX, and LDL contents increased with an increase in altitude, and the increase was greater in the TS45 Tibetan sheep compared those of the TS35 and TS25 sheep (*p* < 0.05) (Table 3).

### 3.2. Analyses of the Structure of the Cardiac Tissues

As shown in Figure 1, the myocardial fibers of the Tibetan sheep at the three altitudes were closely arranged (Figure 1a–c). The number of capillaries per unit area was higher in the TS45 Tibetan sheep than in the TS35 and TS25 sheep (*p* < 0.05) (Figure 1d), and the diameter was mildly increased with an increase in altitude, though it did not, however, reach a significant level (Figure 1e).

### 3.3. Identification of the DEmRNAs in the Hearts of the Tibetan Sheep

Twelve libraries were constructed by mRNA-seq yielding 35,848,724–56,406,366 high-quality reads through quality filtering, with 96.60–97.55% mapped to the sheep reference genome (Appendix A). A total of 575 DEmRNAs (308 up-regulated, 267 down-regulated) were identified in TS35 compared to TS25 (Figure 2, Appendix A). In addition, 688 novel genes were identified in the sequencing data (Appendix A). To validate the accuracy of the sequencing data, 20 mRNAs were randomly selected and detected using the quantitative real-time polymerase chain reaction (qPCR). Our validation results indicated that the qPCR results were consistent with the mRNA-seq results (Appendix A).

### 3.4. Identification of DEmiRNAs in the Hearts of the Tibetan Sheep

There were 12 libraries constructed by miRNA-seq yielding 8,894,072–1,565,091 clean reads through quality filtering, with 75.23–79.31% mapped to the sheep reference genome (Appendix A). A total of 40 DEmiRNAs (27 up-regulated, 13 down-regulated) were identified in the TS35 compared to TS25 (Figure 3, Appendix A). To validate the accuracy of the sequencing data, 10 miRNAs were randomly selected and detected using qPCR. Our verification test indicated the qPCR results to be consistent with the mRNA-seq results (Appendix A).

### 3.5. Functional Analysis of the DEmRNAs

The GO and KEGG were analyzed and the DEmRNAs were found to be mainly involved in the PPAR, AMP-activated protein kinase (AMPK), PI3K-Akt, and cancer-related signal pathway (Figure 4a–c). These DEmRNAs were found to localize in the cell, cell membranes, and organelles, participating in metabolism, biological regulation, and the intracellular process, as well as playing various roles in binding, catalysis, and molecular transmission (Figure 4d). A large number of the DEmRNAs in the three groups were gathered in the metabolic process of biological regulation and organelles and macromolecules in cellular components, indicating that mitochondria and glucose metabolism might play an important role in hypoxia adaptation in Tibetan sheep. The enrichment of the KEGG pathway demonstrated the cancer-related pathways (microRNA in cancer, prostate cancer) to be significantly enriched in TS25-vs-TS35, TS35-vs-TS45 (pathway in cancer), and TS25-vs-TS45 (microRNA in cancer, pathway in cancer) in the first 20 pathways, indicating that the micro-hypoxic environment created by the competitive oxygen absorption of cells during the development of cancer cells is similar to the groups at different altitudes (different oxygen content).

### 3.6. Hypoxia Related DEmRNA–DEmiRNA Association Analysis and Functional Analysis of the Target Genes

To explore the role of mRNAs and miRNAs in hypoxia, the correlation between miRNAs and target genes was evaluated using PCC. The co-expression negative mRNA–miRNA relationship pairs were screened (Appendix A). The GO and KEGG functions of the top 500 mRNAs which had the most significant differences in the network were annotated using the David online annotation tool. The GO (Figure 5A) and KEGG (Figure 5C) analyses identified that DEmRNAs mainly participate in protein phosphorylation, gluconeogenesis in mitochondria through the calcium, Wnt, cAMP, and cancer-related pathways (Appendix A). The co-expression network of DEmRNAs and miRNAs were constructed by selecting 39 mRNAs and 64 miRNAs (Figure 5B, Appendix A). According to the co-expression network, mRNA–miRNA relationship pairs have an important role in mitochondrial dynamics (Figure 5B(a)) and functions (Figure 5B(b)), that is, differentially-expressed mRNAs and miRNAs under hypoxia mainly participate in regulating mitochondrial structure and function to adapt to the hypoxic environment.

### 3.7. qPCR Analysis of the Mitochondrial Dynamic-Related Genes

The expression patterns of the mitochondrial dynamic-related genes in the various tissues of the Tibetan sheep were detected using qPCR (Figure 6). The expression pattern of the heart was considered as an example, where the expression of *Drp1* (Figure 6a), *Fis1* (Figure 6b), and *MFF* (Figure 6c) in the heart was found to be significantly higher in the TS45 sheep compared to the TS25 sheep (*p* < 0.05), the *Drp1* expression was found to be significantly higher in the TS45 sheep than in the TS35 sheep (*p* < 0.05), and the *MFF* expression was found to ve significantly higher in the TS35 sheep than in the TS25 sheep (*p* < 0.05). In addition, the *Mic60* (Figure 6d) and *OPA1* (Figure 6e) expressions were significantly decreased with an increase in altitude, which was lower in all of the tissues of the TS45 and TS35 Tibetan sheep compared to the TS25 sheep (*p* < 0.05).

### 3.8. Immunohistochemical Analysis of Mitochondrial Dynamic-Related Proteins

The expression of proteins presented a similar pattern as the mitochondria-related genes (Figure 7). With the expression pattern of the heart taken as an example, the expression of Drp1 (Figure 7A), Fis1 (Figure 7B), and MFF (Figure 7C) in the heart were significantly higher in the TS45 sheep than in the TS35 and TS25 sheep (*p* < 0.05). The expression of Mic60 (Figure 7D) and OPA1 (Figure 7E) in all of the tissues were found to significantly decrease in the TS45 and TS35 sheep compared to the TS25 sheep (*p* < 0.05), except for in the quadriceps femoris. The gene and proteins results indicated that the fission rate increased in the heart and decrease in the liver, while the fission rate was found to increase first and then decrease in the lung, brain, and quadriceps femoris. Also, the rate of fusion was found to decrease with an increase in altitude.

### 3.9. Ultrastructural Observation of the Mitochondria

To verify the expression of the genes and proteins, the mitochondrial ultrastructure of the five tissues of the Tibetan sheep were observed using a transmission electron microscope (TEM) (Figure 8) and the number of cristae (Figure 9b), the cristae width (Figure 10c), density (Figure 9d), area (Figure 9e), and aspect ratio (Figure 9f) per unit area were measured. Upon observing the mitochondria, the structure of mitochondria in each tissue was found to be clear and the cristae were complete (except in the case of liver tissue). Considering the results of the heart as an example, the number of mitochondria per unit area of the heart was found to be the largest, and the number of single mitochondrial cristae was the largest, increasing with an increase in altitude (Figure 8a–c). The average number of cristae of a single mitochondrion per unit area is usually the largest in the heart; however, it was significantly decreased in the heart with an increase in altitude (*p* < 0.05), and the cristae width, which is normally widest in the heart, was found to decrease in the heart of the TS45 sheep compared to the TS35 and TS25 sheep (Figure 9c). The mitochondria density in the heart was significantly higher in the TS45 sheep than in the TS35 and TS25 sheep (*p* < 0.05) (Figure 9d). Contrary to mitochondrial density, the mitochondria area (Figure 9e) and aspect ratio (Figure 9f) were significantly decreased in the heart in the TS45 sheep compared to the TS35 and TS25 sheep (*p* < 0.05). The number of mitochondria in the heart was increased in all cases, and the area decreased with an increase in altitude, indicating that there was a greater population of smaller mitochondria in the heart at higher altitudes.

### 3.10. Mitochondrial Function Analysis

To measure the changes in mitochondrial function, the key enzymes of the TCA cycle and OXPHOS were detected (Figure 10). Our previous studies found *PDK4* to significantly increase in all of the tissues in the TS45 sheep when compared to the TS35 and TS25 sheep, therefore the citric acid (CA) content was detected and it was found to be significantly lower in all of the tissues of the TS45 sheep compared to that of the TS35 and TS25 sheep (*p* < 0.05) (Figure 10d), indicating that the TCA cycle might be weakened. In addition, the four complexes of OXPHOS were determined in this study. Upon considering the results of the heart as an example, the nicotinamide adenine dinucleotide (reduced state) (NADH) (Figure 10a), the nicotinamide adenine dinucleotide (oxidation state) NAD+ (Figure 10b), ATP (Figure 10g) content, and cytochrome c oxidase (CO) (Figure 10f) activity in the heart were found to be significantly higher in the TS45 sheep than in the TS35 and TS25 sheep (*p* < 0.05). Succinate dehydrogenase (SDH) activity was significantly decreased with an increase in altitude (*p* < 0.05) (Figure 10e). Furthermore, the ratios of NADH/NAD+ in all the tissues significantly increased with an increase in altitude (Figure 10c). The above results indicated that OXPHOS might be up-regulated with an increase in altitude in Tibetan sheep.

## 4. Discussion

### 4.1. The Changes in Blood Indices Promote the Efficiency of Oxygen Transport

This study aimed to study the molecular mechanism of hypoxia adaptation in Tibetan sheep. As such, it might provide a reference for improving the ability of hypoxia adaptation in mammals and for studying the dynamic regulation of mitochondria in the development of hypoxia-related illness. By measuring the physiological and biochemical indices of blood, the PO_2_ and SO_2_ were found to significantly decrease with an increase in altitude, indicating that there might be hypoxia in Tibetan sheep. An increase in HGB and HCT values indicated that there were more red blood cells and hemoglobin in Tibetan sheep at a higher altitude, enhancing their oxygen-carrying capacity and ensuring a normal oxygen supply to the tissue cells that, in turn, alleviate the symptoms of hypoxia [33]. PCO_2_ can evaluate the respiratory acid-base reaction, and the decrease in PCO_2_ indicates that an acceleration in respiratory rate leads to an increase in the elimination of CO_2_, making the blood alkaline. Wang et al. (2019) [16] found the respiratory rate of Tibetan sheep at an altitude of 3500 m to be 49.72 times/min, which was significantly higher than that of Small Tailed Han sheep at an altitude of 1500 m. The PCO_2_ was found to decrease significantly with the increase in altitude, which might be caused by the acceleration of respiratory rate as a result of the increase in altitude, with the decrease in the HCO^3-^ concentration and increase in the BE indicating that the acid-base balance shifted to alkaline with the increase in the respiratory rate. Thus, acid-base regulation is important when it comes to adapting to hypoxia.

Hypoxia can easily cause mild to severe acid-base changes from alkalosis to acidosis, and respiratory alkalosis can reduce the sympathetic nerve tension, slow down the cardiopulmonary hypoxic vasoconstriction and cerebral hypoxic vasodilation, and increase the affinity of hemoglobin to oxygen [34]. However, with the increase in hypoxia stress, the glycolysis process is up-regulated, and the increase in LDH and LDH1 can reduce the alkalinity of the blood. At the same time, the metabolic compensation of CK and CK-MB content can enhance the production of ATP for the purpose of adapting to hypoxic conditions [12,35]. However, the specific bias of the acid-base balance and whether it is beneficial or harmful to the oxygen transport and utilization of the tissues depends on the size and sum of each effect [33]. With an increase in the respiratory rate, the increase in SOD, GPX, and LDL can effectively remove the ROS ensuring blood circulation and normal physiological function of the cells. In addition, hypoxia adaptation critically manifests as a change in the tissue structure, and mammals living at higher altitudes have thicker alveolar septum and denser blood vessels [20]. In our study, the myocardial capillary density (the number of capillaries per unit area) significantly increased with an increase in altitude, increasing the blood flow to strengthen the tissue blood supply and alleviate hypoxia.

### 4.2. Adaptive Regulation of mRNAs and miRNAs under Hypoxia

Genes are important for regulating the phenotype, such as the expression of VEGF and the co-expression of *HIF-1a* and *HIF-1b*, promoting angiogenesis and fibrosis to increase the oxygen supply to the tissue [36]. Research on hypoxia based on the mRNAs/miRNAs mostly focuses on human cancer [37], Tibetan pigs [20], and aquatic organisms [38], while Tibetan sheep are less studied. In this study, the Tibetan sheep in TS25, TS35, and TS45 produced 41.22, 41.32, and 43.15 million clean reads on average, similar to the results of about 47 million clean reads obtained by Lin et al. [39] in goat muscle tissues. Although the RNA sources are different, the similar results indicate that the sequencing results can be used for subsequent research. A total of 18140 mRNAs were obtained by sequencing, and then the DEmRNAs were identified in the three groups and their functions were annotated. In the TS25-vs-TS35 and TS25-vs-TS45 results, the five most significant differentially expressed genes were mainly related to cancer with a role in carcinogenesis (leukemia, gastric cancer, breast cancer, etc.) (*CCKBR*, *GSTT1*, *ARID5B*, *TPT1*, *EXTL1*), antioxidant (*GPX1*, *SELENOW*), angiogenesis (*ANGPTL2*, *PTX3*), and erythrocyte morphology (*SELENOW*). However, the high expression of cancer-related genes does not indicate that the incidence rate of cancer is high or that the cells were at a higher metabolic level due to its high expression to cater to the energy demands of the tissue cells. Wang et al. (2019) [40] found that cancer-related genes and pathways were significantly annotated to meet the rapid growth of velvet antler tissue in the deer research. Meanwhile, tumor suppressor genes were also strongly selected, explaining why cancer and tumor suppressor genes were significantly enriched in our results. In addition, the most significantly differentially expressed genes in the TS35-vs-TS45 results were mainly related to vascular protection (*Nr4a3*, *SCN3B*) and maintaining or changing tissue morphology (*DDIT4L*, *Myh6*), which might be due to the adaptation stage of acute hypoxia stress from 3500 m to 4500 m altitude, similar to the slow downstage of cancer cell proliferation. For example, the expression of *SCN3B* in the TS35 sheep was 10.46 times higher than that in the TS45 sheep, and the lower expression of *SCN3B* can protect the intracranial artery endothelial cells from cardiac atherosclerosis [41]. Appropriate myocardial hypertrophy can enhance cardiac power and accelerate blood flow and tissue blood supply, which is a tissue compensatory change that is made in response to hypoxia stress. As a tissue compensatory change made in response to hypoxia, appropriate myocardial hypertrophy can enhance cardiac power. Studies have identified the expression of *DDIT4L* to promote myocardial hypertrophy in response to stress [42].

GO and KEGG were analyzed for a better understanding of the function of DEmRNAs. The results showed that DEmRNAs were mainly enriched in the hypoxia-related pathways, such as cancer, PI3K-Akt, PPAR, MAPK, calcium ion, camp, and metabolism, and performed binding, catalysis, and functions related to molecular regulation in the cells, organelles, and cell membranes in response to biological processes such as biological regulation and stress. These pathways have been annotated to have important regulatory roles in hypoxia and cancer cell development [43]. It is noteworthy that microRNAs in the cancer pathway are significantly enriched under different hypoxia stresses, which also shows that miRNAs play an important role in the process of hypoxia. miRNAs play a negative role in regulating gene expression by pairing with some bases of the target mRNA for degrading or inhibiting mRNA translation [44]. Therefore, the transcriptome profiles of the heart miRNAs were measured in the Tibetan sheep under different altitudes, and a total of 1046 miRNAs were obtained. To further understand miRNA regulation, the association between DEmiRNAs and DEmRNAs was analyzed and a co-expression network was constructed.

The analysis of mRNA functions in the network revealed that although most of the mRNAs with the strongest correlation with miRNAs were also related to cancer and tumorigenesis, these mRNAs were more specifically related to glucose metabolism and oxidative phosphorylation (*MRPL28*, *ALDH1L2*, *ECI1*, *PDP2*) and mitochondrial dynamics (*TIMM8A*, *MTFP1*, *Fis1*, *ErbB*). For example, as a mitochondrial protein, MRPL28 can be used as a target for developing new drugs for gastric cancer [45]. High expression of *PDP2* can inhibit the activity of pyruvate dehydrogenase (PDH), and then weaken the intensity of using pyruvate in mitochondria to weaken cell respiration and OXPHOS [46]. In this study, there was a decrease in the expression of *PDP2* with the increase in altitude, which might be related to mitochondrial OXPHOS. According to the functional annotation of the target genes in the network, the mRNA mainly located in the mitochondria participates in cancer, Wnt, cAMP, FOXO, ErbB, and calcium regulation signal pathways and plays critical roles in angiogenesis, protein phosphorylation, and mitochondria dynamic regulation. Ludikhuize et al. (2020) [47] found that the FOXO and Wnt signaling pathways play an important role in mitochondrial fission in Lgr5 + columnar cells (CBCS), leading to the differentiation of stem cells into goblet cells. The EGF/ErbB signaling pathway is activated under hypoxia, inducing the phosphorylation of phosphoglycerate kinase 1 (PGK1) S203 and its transfer to mitochondria, which results in the phosphorylation of the pyruvate dehydrogenase kinase, thus enhancing the transfer of pyruvate to mitochondria and the TCA cycle [48]. In this study, the expression of *PGK1* in the TS45 sheep was down-regulated by 1.59 times compared to the TS25 sheep, indicating that the transfer of pyruvate to mitochondria was weakened. From the network diagram constructed, the mitochondria were identified to regulate the dynamics and functions related to adapting to hypoxic conditions. Therefore, dynamic changes in mitochondrial dynamics and function were analyzed under hypoxia.

### 4.3. Mitochondrial Fission Increased and Fusion Decreased under Hypoxia

Mitochondrial dynamics are regulated by genes, the expression of Drp1 and its two receptors Fis1 and MFF promote mitochondrial fission [49], and the expression of OPA1 and Mic60 promotes mitochondrial fusion [16]. The expression of Drp1, Fis1, and MFF showed different expression patterns in various tissues with the increase in altitude, where the expression increased in the heart, decreased in the liver, and increased first and then decrease in the lung, brain, and quadriceps femoris. It could be speculated that an increase in hypoxia stress tends to increase mitochondrial fission in the heart, decrease in the liver, and increases first and then decreases in the lung, brain, and quadriceps femoris. Autophagy might increase with the increase in oxidative stress under hypoxia, and autophagy promotes the recruitment of Drp1 to the mitochondrial membrane to enhance fission and resists stress [50]. Lin et al. (2020) [51] found that in the human hepatoma cells, increased Drp1 expression promotes mitochondrial fission and autophagy to meet the energy demand of cancer cell proliferation under hypoxia. Therefore, the up-regulation of mitochondrial fission might be a regulatory strategy for Tibetan sheep to adapt to hypoxia. The down-regulation of mitochondrial fission in the liver, lung, brain, and quadriceps femoris of the TS45 sheep might be a protective mechanism. Zhang et al. (2018) [52] identified that in human islets β cells, excessive mitochondrial fission increases the death of functional islets β cells and increases the incidence of type 1 and type 2 diabetes. Molina et al. (2009) [53] found that when mitochondrial fission reached a certain stage, *Fis1* expression would decrease and fusion decrease to reduce the excessive consumption of ATP in order to protect the cells. In addition, fission is triggered by fusion, although the increase in the division can enhance the exchange of substances and the production of energy between the cells, and the depolarized mitochondria produced after fission generally do not participate in fusion [54]. In other words, the number of mitochondria participating in fusion decreased with an increase in fission within a certain range. We detected that the expression of OPA1 and mic60 were significantly decreased in the various tissues with the increase in altitude, which indicated that, in contrast to mitochondrial fission, fusion in various tissues might decrease with an increase in altitude.

Mitochondrial fusion is not only affected by fission but is also related to changes in autophagy and membrane potential. Under hypoxia, the risk of mitochondrial damage increases with the enhancement of oxidative stress, with the autophagy receptors recognizing the damaged mitochondria and performing lysosome-dependent degradation, resulting in a decrease in fusion [55]. In addition, fusion might be positively correlated with mitochondrial membrane potential, and the respiration and fusion of mitochondria increases with an increase in mitochondrial membrane potential in pancreatic cancer [56]. Zhao et al. (2018) [57] found that in human uterosacral ligament fibroblasts (hUSLFs), cell viability and mitochondrial membrane potential decreased in hypoxia. Therefore, the decrease in fusion might also be caused by the decrease in membrane potential, but the specific regulatory mechanism is unclear to date.

According to the results of TEM, the mitochondrial density per unit area significantly increased in the heart and liver with the increase in altitude, and increased first and then decreased in the lung, brain, and quadriceps femoris. The area and aspect ratio of the single mitochondrion per unit area was significantly decreased in the heart and liver, and decreased first and then increased in the lung, brain, and quadriceps femoris. This confirmed our conjecture that mitochondria fission in the heart accelerates with an increase in hypoxia stress, and the fusion slowed down so that there were numerous smaller mitochondria in the hearts of the TS45 Tibetan sheep. It should be noted that the number and width of single mitochondrial cristae decreased with the increase in altitude. Mitochondrial cristae are the main site of OXPHOS and most of the related enzymes and proteins gather in this place [58]. Does the decrease in the number and width of the single mitochondrial cristae decrease the oxidative phosphorylation level?

### 4.4. TCA Cycle Is Down-Regulated and OXPHOS Is Up-Regulated under Hypoxia

Our previous study found the synergistic expression of *HIF1A* and *PDK4* to promote glycolysis [12]. As a kinase, PDK4 oxidizes and decarboxylates the pyruvate dehydrogenase to reduce its activity, eventually reducing CA synthesis. The CA content significantly decreased with the increase in altitude in this study, which indicates that the TCA cycle might be weakened under hypoxia. In this study, there was a negative regulation between the *PDK4* expression and CA content, proving that the TCA cycle might be down-regulated under hypoxia, and reducing the TCA cycle in cancer cells to reduce oxygen consumption is also an important regulatory strategy of adapting to hypoxia [59]. OXPHOS mainly depends on the four complexes and ATP synthase of the mitochondrial inner membrane respiratory chain. The change in the OXPHOS rate can be identified by measuring the content/activity of key complexes according to the altitude. Complex 1 (NADH) is the largest in the respiratory chain. In this study, the NADH and NAD+ content significantly increased with an increase in altitude, suggesting that OXPHOS might be up-regulated. Because the redox state of the intracellular NAD is a key factor for regulating glucose metabolism [60], a high NADH/NAD+ content indicates the up-regulation of OXPHOS, and the increase in NAD+ activity can also enhance mitochondrial function and increase ATP production [61]. Complex 2 (SDH) is the second independent entrance for electrons into the respiratory chain. Although SDH itself does not produce protons, it can directly oxidize the succinic acid to transfer electrons to the coenzyme Q [62]. In this study, the activity of SDH decreased with altitude and reached a significant level in the heart, lung, and quadriceps femoris. A decrease in SDH activity under hypoxia reduces the oxidation of the succinic acid, reducing the production of ROS and reducing the potential of the mitochondrial membrane to maintain the mitochondrial function [63].

A decrease in the TCA cycle reduces the production of protons into the mitochondrial membrane gap, which further decreases the membrane potential. Moreover, a decrease in SDH activity might also be related to the weakening of the TCA cycle, which reduces the succinic acid and SDH activity. Complex 4 (CO) is the only tissue-specific complex and is involved in regulating development. There is a concomitant action as a rate-limiting enzyme to regulate the rate of OXPHOS [64]. CO activity was found to significantly increase in all five tissues, which proved that the OXPHOS process was significantly up-regulated under hypoxia. However, the reduction in the number and width of single mitochondrial cristae was identified as to whether it could reduce the OXPHOS level. More mitochondria were speculated to be present in the TS45 Tibetan sheep. Therefore, the number and width of mitochondrial cristae might increase in the whole body, demonstrating that the OXPHOS level has been improved. The ATP content need to increase significantly to meet the energy needs of the tissues and cells and resist stress, but the mechanism of production is not very clear under hypoxia. The ATP content was measured in all the tissues and significantly increased with an increase in the altitude. This production mode might depend on the up-regulation of glycolysis and OXPHOS and the down-regulation of the TCA cycle.

Although the results indicated that changes in mitochondrial dynamics and functions play an important role in a hypoxic environment, the stress caused by the death of the animal might affect the mitochondria, and many anesthetics and toxins are fatal to mitochondria. Therefore, the influence of the external environment on the mitochondria should be reduced as much as possible. In addition, although this study simulates different oxygen contents at different altitudes, the results might not be applicable for other mammals or the development of cancer cells due to species specificity. Therefore, further studies are needed on changes in mitochondria in other mammals, but the regulation of the mitochondria in Tibetan sheep under hypoxia can provide a reference for treating hypoxia-related diseases.

## 5. Conclusions

In summary, the increase in the number of red blood cells and hemoglobin, the density of antioxidants, and myocardial capillaries in the blood of Tibetan sheep might promote efficiency in oxygen transportation and reduce the damage caused by ROS. According to the results of the mRNA-seq and miRNA-seq, Tibetan sheep mainly rely on the PI3K-Akt, Wnt, and PPAR signal pathways to up-regulate oncogenes and tumor suppressors to enhance the cell metabolism, increasing the production of ATP and thereby resisting stress due to hypoxia. These genes mainly fulfil this role by regulating mitochondrial dynamics and functions, such that mitochondrial fission increases, fusion decreases, glycolysis and OXPHOS are up-regulated, and the TCA cycle is down-regulated.

## Figures and Tables

**Figure 1 animals-12-00583-f001:**
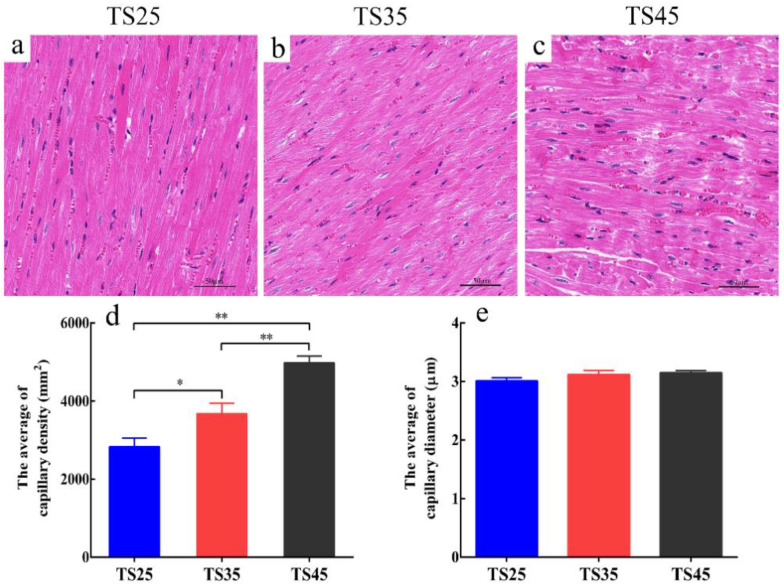
The morphological characteristics of the hearts in Tibetan sheep by H&E staining (40×). (**a**–**c**) Morphological characteristics of heart. (**d**) Capillary density per unit area. (**e**) Average diameter of single capillary per unit area. (**TS25**) 2500 m altitude Tibetan sheep. (**TS35**) 3500 m altitude Tibetan sheep. (**TS45**) 4500 m altitude Tibetan sheep. Data shown on graph are means ± SEM. TS25 as control, * *p* < 0.05, ** *p* < 0.01.

**Figure 2 animals-12-00583-f002:**
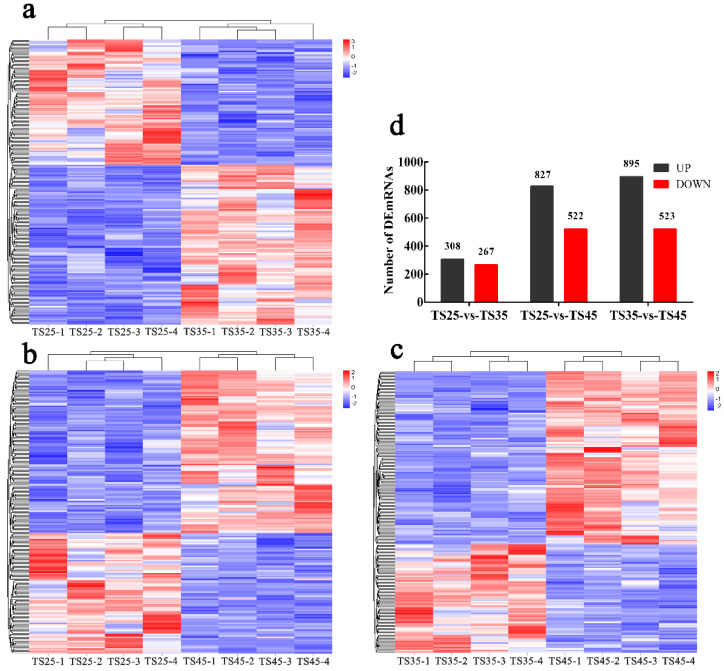
Results of mRNA-seq in heart tissue of Tibetan sheep at different altitudes. The Heatmap shows the relative expression pattern of DEmRNAs between groups. Each column represents a sample, and each row represents the expression of a single mRNA between different samples. The color transitions from blue (low expression) to red (high expression). (**a**) The heatmap between TS25 and TS35. (**b**) The heatmap between TS25 and TS45. (**c**) The heatmap between TS35 and TS45. (**d**) Statistical of DEmRNAs in three group. (**TS25**) 2500 m altitude Tibetan sheep. (**TS35**) 3500 m altitude Tibetan sheep. (**TS45**) 4500 m altitude Tibetan sheep.

**Figure 3 animals-12-00583-f003:**
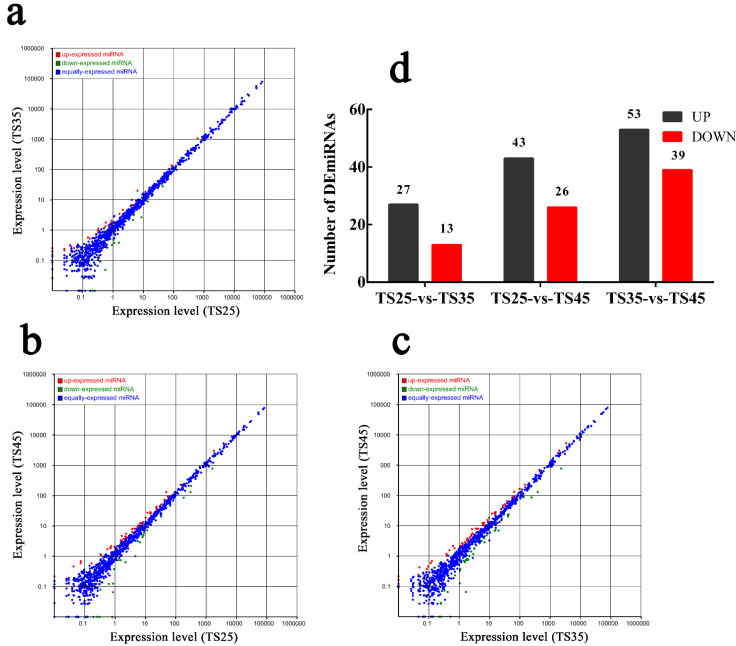
Results of miRNA-Seq in heart tissue of Tibetan sheep at different altitudes. The scatter plot shows that DEmiRNASs were up-regulated and down-regulated between groups. The abscissa is the expression level of the control group, and the ordinate is the expression level of the experimental group. Red indicates up-regulated miRNA, green indicates down-regulated miRNA, and blue indicates co expressed miRNA. (**a**) Scatter plot of TS35 DEmiRNAs relative to TS25. (**b**) Scatter plot of TS45 DEmiRNAs relative to TS25. (**c**) Scatter plot of TS45 DEmiRNAs relative to TS35. (**d**) Statistical of DEmiRNAs in three group. (**TS25**) 2500 m altitude Tibetan sheep. (**TS35**) 3500 m altitude Tibetan sheep. (**TS45**) 4500 m altitude Tibetan sheep.

**Figure 4 animals-12-00583-f004:**
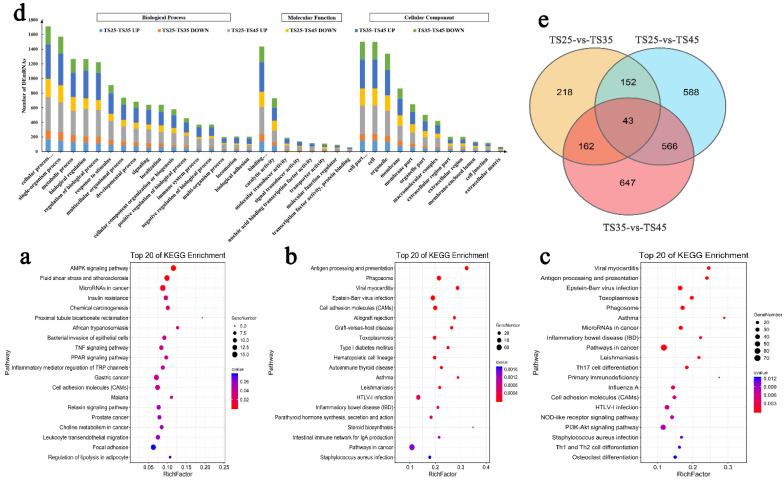
Functional annotation analysis of DEmRNAs in heart tissue of Tibetan sheep at different altitudes. (**a**) Top 20 KEGG enrichment pathways between TS25 and TS35. (**b**) Top 20 KEGG enrichment pathways between TS25 and TS45. (**c**) Top 20 KEGG enrichment pathways between TS35 and TS45. The ordinate is the pathway, and the abscissa is the enrichment factor. Darker colors indicate smaller q-values. (**d**) Histogram of GO annotation results of DEmRNAs. The abscissa is the second level GO term, and the ordinate is the number of DEmRNAs in the term. (**e**) Venn diagram of mRNA interactions based on the overlapping mRNAs among the three groups. (**TS25**) 2500 m altitude Tibetan sheep. (**TS35**) 3500 m altitude Tibetan sheep. (**TS45**) 4500 m altitude Tibetan sheep.

**Figure 5 animals-12-00583-f005:**
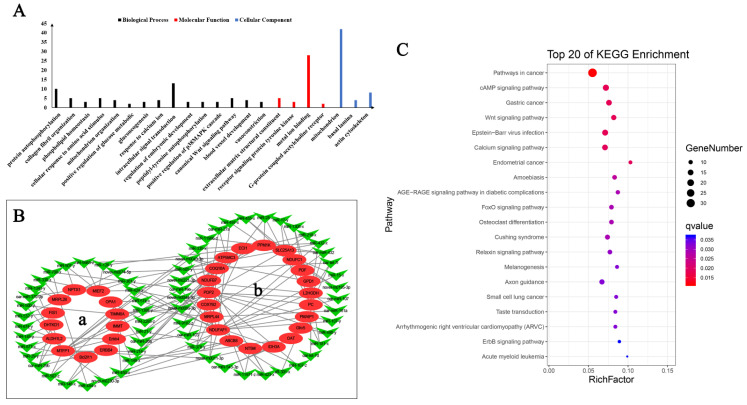
Functional annotation of DEmiRNA target gene and co-expression network. (**A**) Histogram of GO annotation results of top 500 DEmRNAs in mRNA–miRNA correlation analysis results. The abscissa is the second level GO term, and the ordinate is the number of DEmRNAs in the term. (**B**) Gene co-expression network analyses of hypoxic DEmRNAs and DEmiRNAs. (**a**) Mitochondria dynamic related miRNA and target genes. (**b**) Mitochondria function related miRNA and target genes. (**C**) Top 20 KEGG enrichment pathways of top 500 mRNA in mRNA–miRNA correlation analysis results. The ordinate is the pathway, and the abscissa is the enrichment factor.

**Figure 6 animals-12-00583-f006:**
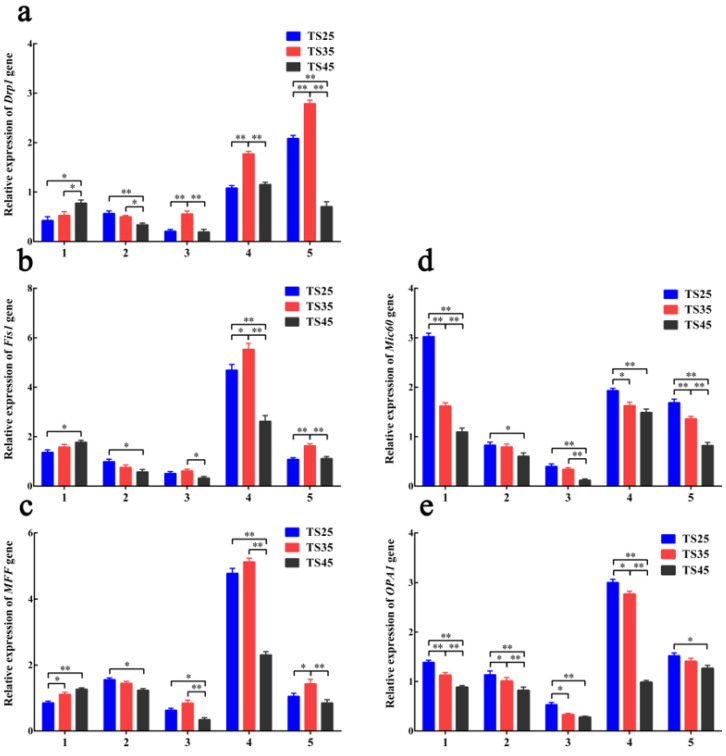
The expression of mitochondria fission (*Drp1* (**a**), *Fis1* (**b**), *MFF* (**c**)) and fusion (*Mic60* (**d**), *OPA1* (**e**)) genes in five tissues of Tibetan sheep. (**1–5**) Heart, liver, lung, brain, quadriceps femoris. (**TS25**) 2500 m altitude Tibetan sheep. (**TS35**) 3500 m altitude Tibetan sheep. (**TS45**) 4500 m altitude Tibetan sheep. Date shown on graph are means ± SEM. * *p* < 0.05, ** *p* < 0.01.

**Figure 7 animals-12-00583-f007:**
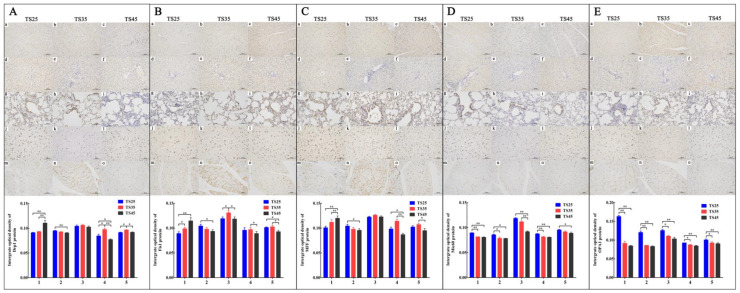
Immunostaining of mitochondria fission (Drp1 (**A**), Fis1 (**B**), MFF (**C**)) and fusion (Mic60 (**D**), OPA1 (**E**)) proteins in five tissues of Tibetan sheep (20×). (**a**–**c**) Heart. (**d**–**f**) Liver. (**g**–**i**) Lung. (**j**–**l**) Brain. (**m**–**o**) Quadriceps femoris. (**1**–**5**) Heart, liver, lung, brain, quadriceps femoris. (**TS25**) 2500 m altitude Tibetan sheep. (**TS35**) 3500 m altitude Tibetan sheep. (**TS45**) 4500 m altitude Tibetan sheep. Date shown on graph are means ± SEM. The IOD in heart of TS25 as control, * *p* < 0.05, ** *p* < 0.01.

**Figure 8 animals-12-00583-f008:**
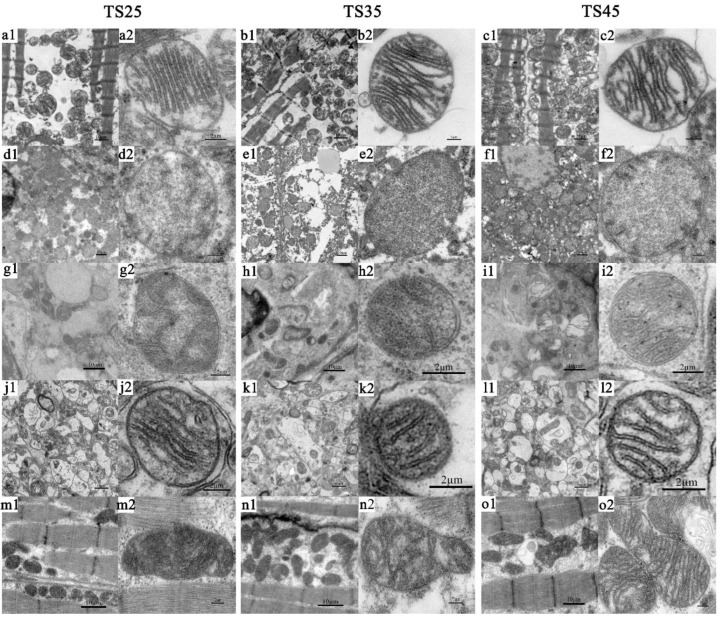
Representative electron micrographs of mitochondria in five tissues of Tibetan sheep. (**a**–**c**) Heart. (**d**–**f**) Liver. (**g**–**i**) Lung. (**j**–**l**) Brain. (**m**–**o**) Quadriceps femoris. (**a1**–**o1**) 2000×. (**a2**–**o2**) 10,000×. (**TS25**) 2500 m altitude Tibetan sheep. (**TS35**) 3500 m altitude Tibetan sheep. (**TS45**) 4500 m altitude Tibetan sheep.

**Figure 9 animals-12-00583-f009:**
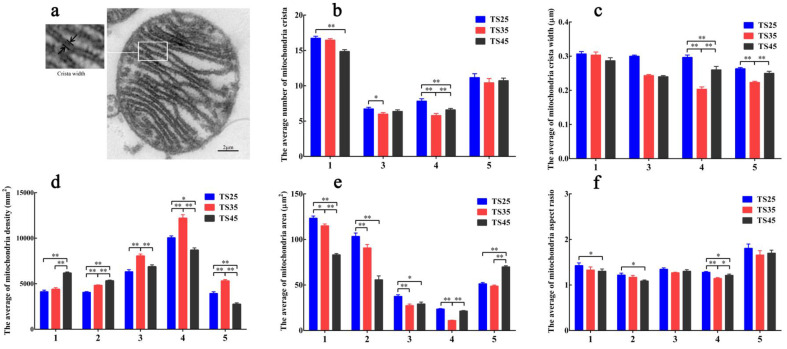
Analysis of five mitochondria dynamic-related indexes in five tissues of Tibetan sheep. (**a**) The structure of the mitochondrial cristae. (**b**) The number of mitochondrial cristae. (**c**) The width of mitochondria cristae. (**d**) Mitochondria density. (**e**) Mitochondria area. (**f**) Aspect ratio. (**1**–**5**) Heart, liver, lung, brain, quadriceps femoris. (**TS25**) 2500 m altitude Tibetan sheep. (**TS35**) 3500 m altitude Tibetan sheep. (**TS45**) 4500 m altitude Tibetan sheep. Date shown on graph are means ± SEM. With the indexes in heart of TS25 as control, * *p* < 0.05, ** *p* < 0.01.

**Figure 10 animals-12-00583-f010:**
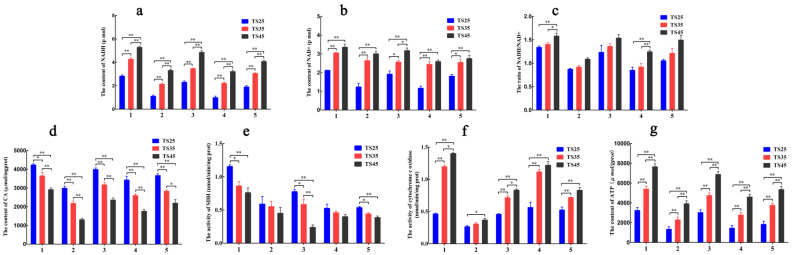
Analysis of mitochondria function-related indexes in five tissues of Tibetan sheep. (**a**) Nicotinamide adenine dinucleotide (NADH (reduced state)). (**b**) Nicotinamide adenine dinucleotide (NAD+ (oxidation state)). (**c**) The ratio of NADH/NAD+. (**d**) Citric acid (CA). (**e**) Succinate dehydrogenase (SDH). (**f**) Cytochrome c oxidase (CO). (**g**) Adenosine triphosphate (ATP). (**1**–**5**) Heart, liver, lung, brain, quadriceps femoris. (**TS25**) 2500 m altitude Tibetan sheep. (**TS35**) 3500 m altitude Tibetan sheep. (**TS45**) 4500 m altitude Tibetan sheep. Date shown on graph are means ± SEM. * *p* < 0.05, ** *p* < 0.01.

**Table 1 animals-12-00583-t001:** The primer information on mitochondria dynamic-related genes.

Gene	GenBank ID	Primer Sequence (5′→3′)	Product Size (bp)	Annealing Temperature/°C	Application
*OPA1*	XM_012140446.1	F: ATGAAATAGAACTCCGAATG R: GTCAACAAGCACCATCCT	112	60	qPCR
*Mic60*	XM_012169573.1	F: TTGAGATGGTCCTTGGTT R: TTGTTTCTGAGGTGGTGAG	136	60	
*Drp1*	XM_015094867.2	F: TCACCCGGAGACCTCTCATT R: TCCATGTAGCAGGGTCATTTTCT	93	60	
*MFF*	XM_027965256.1	F: TCCAGCACGTGCATACTGAG R: CCGCCCCACTCACTAAATGT	107	60	
*Fis1*	XM_027961118.1	F: TGAAGTATGTGCGAGGGCTG R: CCATGCCCACTAGTCCATCTTT	108	60	
*β-actin*	NM_001009784.3	F: GCTGTATTCCCCTCCATCGT R: GGATACCTCTCTTGCTCTGG	97	60	Reference gene
*RPL19*	XM_012186026.3	F: AATGCCAATGCCAACTC R: CCCTTTCGCTACCTATACC	151	60	

**Table 2 animals-12-00583-t002:** Blood physiological indexes of Tibetan sheep at different altitudes.

Blood Physiological Indexes	Tibetan Sheep		
	TS25	TS35	TS45
Oxygen pressure, PO_2_ (mmHg)	37.75 ± 2.02 ^a^	34.00 ± 1.78 ^a^	28.50 ± 1.04 ^b^
Oxygen saturation, SO_2_ (%)	72.00 ± 0.82 ^a^	65.75 ± 0.91 ^b^	59.75 ± 0.83 ^c^
Hemoglobin, HGB (g/dL)	11.60 ± 0.44 ^c^	13.05 ± 0.44 ^b^	15.80 ± 0.37 ^a^
Hematocrit, HCT (%PCV)	34.75 ± 0.85 ^c^	39.50 ± 0.87 ^b^	50.50 ± 0.87 ^a^
Potential of hydrogen, PH	7.37 ± 0.03 ^a^	7.33 ± 0.02 ^a^	7.31 ± 0.01 ^a^
Carbon dioxide pressure, PCO_2_ (mmHg)	55.70 ± 1.42 ^a^	40.75 ± 1.19 ^b^	34.92 ± 1.64 ^c^
Concentration of bicarbonate, HCO^3-^ (mmol/L)	27.35 ± 0.79 ^a^	25.38 ± 0.83 ^a^	25.03 ± 0.81 ^a^
Base excess, BE (mmol/L)	3.00 ± 0.41 ^a^	3.25 ± 0.48 ^a^	3.75 ± 0.48 ^a^

Date shown on table are means ± SEM, and different lowercase letters indicate that the difference was significant.

**Table 3 animals-12-00583-t003:** Blood biochemical indexes of Tibetan sheep at different altitudes.

Blood Biochemical Indexes	Tibetan Sheep		
	TS25	TS35	TS45
Creatine Kinase, CK (U/L)	256.60 ± 4.43 ^c^	300.34 ± 4.31 ^b^	499.62 ± 6.38 ^a^
Creatine kinase isoenzymes, CK-MB (U/L)	42.12 ± 1.86 ^c^	49.70 ± 1.90 ^b^	58.91 ± 2.12 ^a^
Lactate dehydrogenase, LDH (U/L)	616.57 ± 7.68 ^c^	833.31 ± 6.37 ^b^	906.08 ± 5.09 ^a^
Lactate dehydrogenase isoenzymes, LDH1 (U/L)	131.34 ± 4.25 ^b^	156.68 ± 4.48 ^a^	164.78 ± 5.23 ^a^
Superoxide dismutase, SOD (U/mL)	198.03 ± 4.77 ^b^	206.28 ± 4.59 ^b^	244.20 ± 4.65 ^a^
Glutathione peroxidase, GPX (U/mL)	54.76 ± 3.03 ^b^	64.83 ± 2.46a ^b^	69.06 ± 5.49 ^a^
Low-density lipoprotein, LDL (mmol/L)	0.52 ± 0.06 ^b^	0.72 ± 0.04 ^a^	0.81 ± 0.03 ^a^

Date shown on table are means ± SEM, and different lowercase letters indicate that the difference was significant.

## Data Availability

The data presented in this study is contained within the article.

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
