# Peer review of "Changes in the Mitochondrial Dynamics and Functions Together with the mRNA/miRNA Network in the Heart Tissue Contribute to Hypoxia Adaptation in Tibetan Sheep"

_animals, 2022, doi:10.3390/ani12050583_

Round 1

Reviewer 1 Report

Summary:

In this manuscript, Wen and co-workers study the effects of hypoxia on blood oxygenation markers, gene expression profile, metabolic state of cells and mitochondrial morphology of sheep at three different high altitudes. The authors find that, indeed, higher altitude is associated with hypoxia-induced differential expression of several genes associated with cancer, vascular development and metabolism. These expression changes lead to alterations in mitochondrial morphology, energy production and are claimed to be an adaptive mechanism by the animals to resist hypoxic stress.

Overall, the experiments appear well-performed, the differences are robust and a lot of data has been deposited to a public database, which may drive future data-mining research. The work provides some interesting insights into the adaptive mechanisms of mammals to high altitudes and may have implications for understanding oxidative stress in cancer. Nevertheless, prior to publication, I have some concerns that should be addressed to improve the manuscript, as detailed below:

Major Concerns:

It is unclear how controlled these experiments are; while samples from sheep/ewes were taken at different altitudes, how do the authors know that the sheep have only been maintained at the altitude that they have collected samples?

Section 3.6 – miRNAs can target more than one gene, yet the authors only chose 39 mRNAs to correlate. It is unclear why the number of mRNAs in the correlation is less than the number miRNAs.

Fig. 8 – The authors show some statistically significant differences between the TS25 and TS35 groups as well as the TS35 and TS45 groups, yet these differences are not discussed in Section 3.7.

Fig. 9 – While the IHC staining strengthens the paper by providing expression level data at the protein level, western blotting would be even more convincing. Do the authors have any western blot data to complement the IHC?

Fig 11 – Why do panels b and c contain only 4 groups? What happened to group 2?

Section 3.10 – Why would the TCA cycle be weakened (line 456), yet the ATP, NAD+, cytochrome C, etc.. and other markers of OXPHOS activity be higher in TS45 compared to TS35 and TS25. More clarification for this argument is needed.

Section 4.1 – If Hb and the oxygen carrying capacity of sheep at high altitudes is upregulated, why would there be decreased PO2 and hypoxia?

Lines 523-526 – when the authors mention genes “related to cancer”, do they mean “oncogenes”. The terminology needs to be clarified and more explanation as to what precisely is mean by “related to cancer” or “cancer-related genes”.

If cancer-related genes are upregulated in high-altitude sheep, why is there not a higher incidence of cancer in these animals?

Lines 632-635 – I don’t quite understand how mitochondrial fission in the heart would lead to more and “larger mitochondria”; wouldn’t this lead to more mitochondria only? The entire discussion on fusion and fission is convoluted and difficult to follow. I would suggest extensive clarification and simplification of this part of the discussion.

Lines 663 – It is not clear how a reduced mitochondrial membrane potential would maintain mitochondrial function; more explanation on this mechanism is needed.

Minor Concerns:

Introduction should be separated out into multiple paragraphs rather than large blocks of text. This will make the intro easier to read.

Table 6 and 7 could probably be moved to the Supplementary Info.

The fonts in Fig. 2/3/4/5/6/7/8/9/10/11 are too small to read. The font sizes need to be dramatically increased.

Lines 506-508, it would be good for the reader if “capillary density” is defined again at this point of the manuscript.

Discussion – the large blocks of text need to be broken up into smaller paragraphs, so that the manuscript is easier to read

Author Response

Dear Reviewer 

Thank you for the opportunity to revise this manuscript. Your comments has been adopted and gave a point-to-point reply . Please check the manuscript for specific revision details (revised manuscript-marked), and check the file “Response to reviewer” for detailed responses to all questions. 

Kind regards

Yuliang Wen

Reviewer 2 Report

The authors have taken a novel approach to examine the effects of hypoxia on several parameters, but mainly focussing mitochondrial function by studying the effects of reduced oxygen partial pressure associated with increasing altitudes. Their model was to take samples from Tibetan sheep kept at increasing altitudes and look at a range of biomarkers taken from blood, as well as tissues, and in the latter case, also so some EM to look at organelle structure. The concept, and the data itself is fairly unique and very worthy of publication, unfortunately, the paper itself still requires a lot more work. This is mainly because it is unfocussed and far too speculative, and does not cite the broader literature in this field adequately; this could alter the authors conclusions:

  • The authors do not cite a lot of the literature on what is already known about high altitude effects, some of which contradicts their findings (e.g., a reduction in mitochondrial number and capacity, which has often been observed, and effects on mtDNA copy number for instance). There are plenty of good reviews out there, such as those by Murray 2016, Steinen 2020, Wang 2021, Witt 2019 & Storz 2021. They need to discuss this more broadly in both the introduction and the discussion. The key point here is that hypoxia, especially with height, places a much greater strain on the system, as it often also requires the ETC to also generate heat (so get into things like VO2 max). Somewhat paradoxically, many species seem to have a mild reduction in mitochondrial function, but this could be coupled to enhanced respiratory ratio: this could be an adaption to oxidative stress. The mechanism here is still not fully clear and is likely to vary between different species.
  • The authors greatly over-interpret their findings, and in some cases, a deeper understanding of mitochondrial function would really help, in particular, in cancer and how the Hif pathway is involved. The main point is that mitochondrial function changes in cancer, but is still very important, and the TCA also changes function – so the concept of greater or reduced is perhaps not really applicable. Plus, hypoxia is often not important especially in non-solid tumours, but the pathways themselves are, as they are involved, for instance, in the Warburg shift.
  • There are also alternate electron acceptors, such as fumarate. Plus, what is the role of the PPP and NADPH? The NADH/NAD+ ratio can be very misleading, as it could simply represent over-reduction of the ETC. Is the NADH predominantly mitochondrial? What about sirtuins, what would be predicted? Why does SDH change?
  • What is the relationship of the changes they observe to inflammation?
  • Need to explain more about the origins of the sheep. Have they had time to adapt over several hundred years, like their human herders, or have they only recently moved up, or down, the mountains? Big difference between evolved (genetic) adaptation and acute adaptation (e.g., epigenetic). Need to make this clear.
  • How, and where, were the sheep euthanised? Did they spend their entire lives at one altitude, or were they moved around? Stress could have a massive impact on mitochondrial phenotype. Must remember that death by anaesthetic vs, say, shooting or bloodletting could result in big differences; many organs do not die straight away, and their cells can continue to function, but under increasing stress. Many anaesthetics are mitochondrial toxins at high doses.
  • Mitochondrial dynamics. Changes in the known genes involved in fission and fusion, even their expression, is not always indicative of the physiological mitochondrial structure as the process is controlled on so many levels. The only real way to study this is in live cells using mitochondrial dyes, or autofluorescence. Even EM micrographs can be misleading. A major point here is that different cells type display different fusion/fission balances, but in general, fusion is a way to enhance ox phos, and is well described as a stress response. Fission can occur at very high stress levels and can be linked into mitophagy and a switch to glycolysis. This is controlled by both mitochondrial membrane potential and production of ROS, but can be suppressed by, for instance, changes in inflammation and control of calcium flux. One of the key findings relating to the Warburg effect in cancer is that not only is key in provided metabolites for growth, but it also has profound effects on apoptosis. The observations about cristae, are however, if real, very important. (See paper by Klecker & Westermann, 2021)
  • Studying mitochondrial function post-mortem is very difficult and open to interpretation; need to make this point.
  • In terms of hb, evidence is that the concentration, and haematocrit, eventually falls, but the volume of blood increases as an adaptive measure. Again, this could be to compensate for oxidative stress. (The authors mention sheep and water, but a common finding, at least in acute adaptation, is dehydration.)
  • Mitochondrial distribution is altered in some cells, suggestive of diffusion modulation. Have to remember that oxygen is toxic, so its diffusion is very carefully controlled.
  • What is AMS and its relevance (line 135)?
  • How much tissue was used to extract the RNA?
  • In the copy of the paper under review, many of the figures were too small and the font was almost unreadable; need to redesign the figures and legends to make them easier to read.
  • What about fatty acid oxidation – would PPAR alpha expression increase or decrease? Data suggest that in high altitude adapted animals, their capacity for FAO decreases.

Suggestions summary:

  • Need to focus the paper more on what was actually found, and greatly reduce the speculation (so simplify); although the link to cancer is an interesting approach, the authors either need to provide a more concise rationale, or perhaps to focus on the high altitude aspects and what can be said without trying to over-interpret
  • In both the introduction and discussion, the authors need to demonstrate a better understanding of the high-altitude literature, as well as basic mitochondrial function and then re-evaluate their conclusions.
  • Get a good English editor to help rewrite. There are a lot of very basic grammatical errors.

Author Response

(The authors gave the same response as above.)

Reviewer 3 Report

This study aims to study the molecular mechanism of hypoxia adaptation in Tibetan sheep, which may provide a reference for improving the ability of hypoxia adaptation in mammals and studying the dynamic regulation of mitochondria in the development of cancer cells. This study was well organized. Data are sound, however, there are still have several issues that need to be addressed. A revision is suggested.

  1. Figures are too small to read. Please improve.
  2. H&E staining in Fig1 is unclear.
  3. Inconsistent scale bars were presented.
  4. Mitochondrial biogenesis should be investigated.
  5. Please discuss the limitation of this study.

Author Response

(The authors gave the same response as above.)

Reviewer 4 Report

Mitochondrial dynamic and function changes together with mRNA/miRNA network in heart tissue contribute to hypoxia adaptation of Tibetan sheep

Wen, et. al., Animals

The authors present a study of the effects of differing altitudes on the transcriptomic profiles and metabolic dynamics of various tissues of Tibetan sheep.  The results are of general interest as hypoxic conditions are known to alter metabolic function.  Additionally, the experiments performed are sufficient for a thorough characterization of these sheep.  The primary hypothesis tested is that hypoxia at increasing altitudes can alter the metabolic function of these tissues.  Though the study is meritorious, the manuscript was hastily prepared and requires major modifications prior to being suitable for publication.  Specifically, the authors distract from the primary (metabolic) characterization of the paper by discussing relevance to cancer.  Additionally, several of the figures/tables are not presented in an appropriate fashion and virtually all figures are difficult to read at their current size.  Indeed, some are not even possible to read and thus cannot be evaluated at present.  Finally, careful attention needs to be paid to English grammar.  Several other specific suggestions are listed below.

Abstract – The abstract is quite long.  Twice as long, in fact, than most standard abstracts of < 250 words.  For conciseness, it is highly recommended that the abstract be thinned out to remove very specific findings (e.g. lines 33-40 contain very specific expression information that may not be necessary in the abstract)

Line 58 – Aerobic metabolism encompasses all mitochondrial respiration, whether glucose, lipids, or amino acids are used as metabolites.  Please remove the word “glucose”.

Lines 78-85 should be a new paragraph

Lines 91-103 should be a separate paragraph

Lines 52 and lines 103-112 – The mention of cancer and cancer treatment seems to distract from the primary purpose of this article.  Studying mitochondrial dynamics in the presence of varying oxygen levels itself is meritorious and of general interest to the field.  The discussions of cancer could be removed in order to streamline the focus of the manuscript

Lines 112-120 should be a separate paragraph

Statistical methods – There is no mention of a statistical power analysis to determine in 4 animals per altitude would power the study sufficiently.  Please at least include a post-hoc power analysis to show whether the study was appropriately powered.

Tables 1&2 – These tables are absolutely essential for transparent methods reporting.  However, it is recommended that these tables be put into the supplementary files for a more focused manuscript.

Tables 4&5 – some of the letters are written in superscript, and some are not.  Please fix this.  Also, it is unclear from the figure legends how to interpret what an “a” versus a “b” or a “c” indicates.  Please explain what each letter means distinct from the others in the legends.

Figure 9 is impossible to see even at the highest magnification.  A figure of with this much data could easily take up an entire page if shown adequately.  At present, none of the labeled are legible even at 5x magnification.

Figure 10 is not properly labeled so as show which electron micrographs are from 25, 35, and 45 Km sheep.

Lines 523-539 and throughout the discussion: Again, this discussion of the effects of hypoxia on cancer does not seem relevant to this manuscript and should be removed for a more streamlined focus.

Author Response

(The authors gave the same response as above.)

Reviewer 5 Report

Major concerns

  1. Grammars and writing should be checked thoroughly and corrected. Here are some examples in the Abstract.

Ex. Title: Mitochondrial …..mRNA/miRNA network in the heart tissue contribute to hypoxia adaptation of Tibetan sheep

Ex, line 13-14, Long-term exposure to hypoxia, one of the important cellular stresses, can induce hypoxia-related diseases and even death.

Ex, line 16,-17, in the Qinghai-Tibet Plateau, where they have been inhabited and  well adapted to the plateau hypoxia.

EX, line 19, altitudes.

EX, line 17-21, In this work, a systematic analysis including the blood indexes, tissue morphology, mRNA and miRNA expression regulation and mitochondrial function changes of Tibetan sheep at different altitudes, was carried out to provide insights for mechanism of animal hypoxia adaptation and the progression of hypoxia-related illnesses.

EX, line 22-26, In order to explore the adaptation mechanism to hypoxia, the study systematically analyzed blood indexes, tissue morphology, mRNA/miRNA regulation, mitochondrial dynamics and functional changes in Tibetan sheep raised at different altitudes to provide insights for molecular regulation and mitochondrial functionality under hypoxia.

Ex. 26-28, In blood indexes and myocardial morphology, HGB, HCT, CK, 26 CK-MB, LDH, LDH1, SOD, GPX, LDL level, and myocardial capillary density were significantly increased in sheep at higher altitudes (P < 0.05).

EX, line 28-31, RNA-Seq results suggested that DEmRNAs and DEmiRNAs were mainly associated with PI3K-Akt, Wnt, and PPAR signaling pathways in accompany with up-regulation of oncogenes (CCKBR, GSTT1, ARID5B) and tumor suppressor factors (TPT1, EXTL1, ITPRIP) to enhance cellular metabolism and increase ATP production.

EX, line 32-43, enriched; for example, MTFP1 as….and NDUFAF1 OXPHOS regulation ,….

EX, line 35-37, Mitochondrial dynamics, the expression ….were increased with increasing altitudes (P<0.05).

EX, line 39-42, The density of mitochondria was s….. altitudes (P < 0.05).

Ex. line 42-44, The NADH, NAD+ and ….were significantly increased, whereas SDH and CA activity were significantly decreased in various tissue with increasing altitudes (P < 0.05).

Ex. line44-46, Accordingly, the changes of blood indexes and myocardial morphology in Tibetan sheep would improve the efficiency of hemoglobin carrying oxygen and reduce oxidative stress in in Tibetan sheep. The higher expression of oncogenes and tumor suppressor factors may facilitate cell division and energy exchange as evidenced by enhanced mitochondrial fission and OXPHOS expression but reduced fusion and TCA cycle for more and rapid production of ATP in adaption to hypoxia stress.

Ex. line 49-52, For the first time, this systematic study delineated the mechanism of hypoxia adaptation in the heart of Tibetan sheep.

  1. Too long and tedious in the abstract.
  2. At the end of abstract, “studying the dynamic regulation of mitochondria during the development of cancer cells”. How the authors concluded the tissues/organs develop into cancers? The authors lacked obvious evidences to make the conclusion. Totally, postulations.
  3. The strain of sheep should be described.
  4. The clone (access number in Genbank) for primers used for qRT-PCR should be noted.
  5. The resolution of images and histograms should be enlarged.

Author Response

(The authors gave the same response as above.)

Round 2

Reviewer 3 Report

My questions had been well addressed, this submission is acceptable

Author Response

Dear Reviewer 
Thank you again for the opportunity to revise this manuscript. Your comments has been adopted and gave a point-to-point reply . Please check the manuscript for specific revision details (revised manuscript-marked), and check the file “Response to reviewer” for detailed responses to all questions. 
Kind regards
Yuliang Wen

Reviewer 4 Report

The authors have addressed most of the original concerns.  However, several issues remain that still need to be addressed.  For Example:

1)  Power analysis is a statistical calculation that ensures that group sizes are adequately large for detecting the reported differences.  Generally this is done before the study takes place.  Please consult with a statistician on performing this calculation now that the study is complete.

2) Figures 10 and 13 are still too grainy to read, even at high magnification.  This must be corrected prior to publication.

Author Response

(The authors gave the same response as above.)

Reviewer 5 Report

Writings and grammars require moderate revision and correction, such "causes" in line 55 

Author Response

(The authors gave the same response as above.)
